# Development of a Chicken Immunoglobulin Heavy Chain Variable Region (VH) Single-Domain Antibody (sdAb) Against Calsequestrin (CSQ) and Its Application

**DOI:** 10.3390/antib14030080

**Published:** 2025-09-19

**Authors:** Sun Lee, Seoryeong Park, Hyunji Yang, Geummi Cho, Seung Youn Lee, Donggeun Lee, Nara Tae, Dae Hee Kim, Junho Chung

**Affiliations:** 1Department of Biochemistry and Molecular Biology, Seoul National University School of Medicine, Seoul 03080, Republic of Korea; sunlee90@snu.ac.kr (S.L.); psr9805@snu.ac.kr (S.P.); jk05025@snu.ac.kr (H.Y.); aumi0715@snu.ac.kr (G.C.); younuou@snu.ac.kr (S.Y.L.); donggeun.lee98@gmail.com (D.L.); 2Cancer Research Institute, Seoul National University School of Medicine, Samsung Cancer Research Building, 514, 101 Daehakro, Jongno-gu, Seoul 03080, Republic of Korea; 3Institute on Aging, Seoul National University School of Medicine, Seoul 03080, Republic of Korea; 4Department of Biomedical Sciences, Seoul National University School of Medicine, Samsung Cancer Research Building, 514, 101 Daehakro, Jongno-gu, Seoul 03080, Republic of Korea; 5Kangwon Institute of Inclusive Technology, Kangwon National University, Chuncheon 24341, Republic of Korea; nalgoon@gmail.com (N.T.); kimdh@kangwon.ac.kr (D.H.K.); 6College of Pharmacy, Kangwon National University, Chuncheon 24341, Republic of Korea

**Keywords:** single-domain antibody, chicken-derived VH antibody, Calsequestrin, phage display, antibody engineering

## Abstract

**Background/Objectives:** Calsequestrin (CSQ) is a calcium-binding protein that is highly soluble and can serve as a solubility-enhancing fusion tag in recombinant protein expression. Its unique property of calcium-induced precipitation followed by EDTA-mediated resolubilization enables efficient purification. However, the broader application of CSQ-tagged proteins in research have been hampered by the lack of reliable anti-CSQ detection reagents. This study aimed to develop single-domain antibodies (sdAbs) against CSQ for use in diverse immunoassays and cell-based analyses. **Methods:** Single-domain antibodies were selected from phage-displayed chicken VH libraries generated from CSQ-immunized chickens. After biopanning, CSQ-specific VH sdAb clones were isolated and expressed as VH–human kappa light chain constant region (VH-Cκ) fusion proteins in E. coli. The PE06 clone was chosen for further characterization and conjugated to horseradish peroxidase (HRP) and Alexa Fluor 647 for assay applications. **Results:** PE06 VH-Cκ fusion protein demonstrated specific binding to CSQ-tagged proteins and enabled reliable detection in enzyme-linked immunosorbent assay (ELISA), immunoblotting, and flow cytometry. These results validated its utility as a chemically defined detection reagent for CSQ fusion proteins expressed in E. coli. **Conclusions:** This study establishes a CSQ-specific chicken VH sdAb as a versatile detection tool for CSQ-tagged proteins. The approach expands the utility of CSQ as a protein fusion tag and enables the development of recombinant antibodies fused with CSQ, such as scFv-CSQ constructs, for broad application in research and assay systems.

## 1. Introduction

Calsequestrin (CSQ) is a calcium-binding protein predominantly located in the sarcoplasmic reticulum (SR) of muscle cells, where it is essential for calcium storage and release during muscle contraction [1]. It exists in two primary isoforms: CASQ1, predominantly found in skeletal muscle, and CASQ2, which is expressed in cardiac muscle [2]. Both isoforms are characterized by their ability to bind large quantities of calcium ions with low affinity and high capacity—a feature that allows for efficient calcium buffering within the SR while maintaining a reservoir for rapid calcium release during muscle contraction [2,3].

Structurally, CSQ can adapt its solubility and polymerization behavior in response to calcium levels [4]. Under low-calcium conditions, CSQ exists primarily as a highly soluble monomer, with negatively charged acidic residues—rich in aspartic and glutamic acids—promoting its even distribution within the SR [4,5]. As calcium levels increase, calcium ions bind to these acidic residues, inducing conformational changes that stabilize CSQ and promote self-association. This process begins with the formation of dimers, which further assemble into larger polymeric structures, facilitated by calcium ions acting as bridging elements between negatively charged regions of adjacent molecules [6,7].

The resulting calcium-bound CSQ polymers exhibit reduced solubility, which enhances their capacity to buffer large amounts of calcium efficiently. By transitioning into this more compact and organized state, CSQ prevents calcium precipitation while ensuring that an adequate reservoir of calcium is maintained. During muscle excitation-contraction coupling, stored calcium is rapidly mobilized and released through ryanodine receptors located on the SR membrane, underscoring CSQ’s critical role as a dynamic and efficient calcium reservoir in muscle cells [8].

CSQ has also been found to enhance the solubility of aggregation-prone proteins when expressed as fusion proteins in bacterial and cell-free systems, by mitigating inclusion body formation and improving protein folding efficiency [9]. Moreover, such fusion proteins can be isolated from solution by forming complexes with calcium to precipitate, and subsequently re-dissolved by adding ethylenediaminetetraacetic acid (EDTA)—a process that can serve as a rapid purification step [10]. This simple purification method provides a significant advantage over other conventional fusion tags such as maltose-binding protein (MBP), small ubiquitin-like modifier (SUMO), and glutathion S-transferase (GST), which require affinity-based purification. To enable the use of CSQ-fusion proteins in various experimental applications such as enzyme-linked im-immunosorbent assay (ELISA), immunoblotting, and flow cytometry, the availability of anti-CSQ antibodies would be beneficial, as it would eliminate the need for incorporating additional peptide tags into the fusion protein.

In species such as sharks, camelids, and llamas, unique heavy-chain-only antibodies that naturally lack light chains have been identified. These antibodies function effectively without light chains due to their specialized variable domains. When expressed independently, this variable domain is known as a single-domain antibody (sdAb) or nanobody [11,12]. Unlike conventional antibodies composed of both HCs and LCs, sdAbs are small and agile, allowing them to access epitopes that bulkier classical antibodies cannot reach [13]. Despite their small size, sdAbs retain high specificity and affinity. Moreover, they offer advantages over traditional IgG antibodies, such as efficient production in microbial expression systems, improved solubility, thermal stability, and superior tissue penetration. These properties make them valuable for research, diagnostic, and therapeutic purposes [11,14,15,16].

More recently, birds—particularly chickens—have emerged as promising hosts for antibody generation for several reasons. First, the considerable phylogenetic distance between chickens and humans allows conserved human proteins that are weakly immunogenic in mammals (e.g., mice or rabbits) to elicit robust immune responses in chickens. In addition, similar to camelid variable heavy domain of heavy chains (VHHs), chicken antibodies often have longer complementarity-determining region 3 (CDR3), with an average length of 20 amino acids, which can form deep binding pockets [15]. By comparison, average CDR3 lengths are shorter in mice (11 amino acids), rats (12.1 amino acids), and rabbits (13 amino acids) [17,18]. The longer CDR3 regions in chickens arise from their unique V(D)J recombination mechanism and extensive somatic hypermutation, enabling recognition of antigens that are often difficult for conventional antibodies to target [19].

Finally, and most importantly, unlike mammals, chickens lack distinct kappa (κ) and lambda (λ) LC loci. Instead, they possess a single LC gene locus that is lambda-like in both structure and function [20]. Consequently, the heavy-chain variable region (VH) of chicken immunoglobulin tends to be more structurally autonomous and capable of folding and maintaining stability without requiring a specific variable light chain partner [21,22]. Indeed, it has been reported that chickens with a truncated light chain transgene can express heavy-chain-only antibodies [23]. Furthermore, immunization with a variety of protein antigens rapidly produces high and specific serum titers in these chickens, and high-affinity monoclonal antibodies can be efficiently recovered by single B cell screening. These observations raise the possibility of selecting sdAbs from chicken antibody display libraries composed solely of VH domains.

In this study, we prepared recombinant CSQ protein and immunized chickens to generate a chicken sdAb immune library displaying VH domains. From this library, we successfully selected anti-CSQ chicken sdAbs. These sdAbs were expressed as soluble fusion proteins with the human constant κ light chain domain (Cκ) in a prokaryotic expression system. Thereafter, we evaluated the performance of anti-CSQ sdAb-Cκ fusion proteins in ELISA and immunoblot assays following conjugation with horseradish peroxidase (HRP), and in flow cytometry assays following conjugation with a fluorescent dye.

## 2. Materials and Methods

### 2.1. Cell Culture

All cell lines, including Human Embryonic Kidney 293 (HEK293) cells (Korean Cell Line Bank, Seoul, Republic of Korea), were cultured in Dulbecco’s Modified Eagle Medium (DMEM; Welgene Inc., Gyeongsan-si, Republic of Korea) supplemented with 10% (*v*/*v*) fetal bovine serum (FBS; Gibco, Paisley, UK) and 1% (*v*/*v*) penicillin/streptomycin (Gibco, UK), according to the provider’s recommendations. HEK293 cells overexpressing human claudin 18.2 (HEK293 hCLDN18.2 OE) were maintained in DMEM containing 10% (*v*/*v*) FBS and 400 μg/mL G418 (Gibco, UK) for selection. The human gastric carcinoma cell line SNU-601 was cultured in RPMI-1640 (Welgene Inc., Daegu, Republic of Korea) with supplements identical to those used for DMEM. All cells were subcultured every 48 h at 3 × 10^5^ cells/mL and incubated at 37 °C in a humidified atmosphere containing 5% CO_2_.

### 2.2. Expression and Purification of Recombinant Proteins

The expression and purification of recombinant CSQ fusion proteins were performed as described previously [10]. Briefly, genes encoding NUPR1a-His-HA-CSQ fusion protein (Appendix A), an anti-SARS-CoV-2 VH-His-HA-CSQ fusion protein [24] (Appendix A), and an anti-human claudin 18.2 VHH-His-HA-CSQ fusion protein (unpublished data) were chemically synthesized (IDT DNA, Coralville, IA, USA), cloned into the pET-22b expression vector (Novagen, Madison, WI, USA), and transformed into *E. coli* BL21 (DE3) (New England Biolabs, Ipswich, MA, USA). A single colony was grown in 20 mL Super Broth (SB) medium containing 100 μg/mL carbenicillin until the OD_600_ reached 0.7. 1 mM Isopropyl β-D-1-thiogalactopyranoside (IPTG) was then added, and the culture was incubated overnight at 18 °C. Cells were harvested (6000 rpm, 20 min, 4 °C), resuspended in TSE buffer (200 mM Tris-HCl, pH 8.0, 20% sucrose, 1 mM EDTA), and incubated on ice for 30 min. An equal volume of ice-cold water was added, followed by another 30 min incubation on ice, and centrifugation (8000 rpm, 20 min, 4 °C). The supernatant was supplemented with 20 mM CaCl_2_ and stirred at 4 °C for 1 h. Precipitated protein was collected by centrifugation (5000 rpm, 30 min, 4 °C) and resuspended in EDTA buffer (20 mM sodium phosphate, 0.5 M NaCl, 50 mM EDTA, pH 7.4).

For expression of the CSQ-human Fc (CSQ-hFc) fusion protein, the genes encoding CSQ-hFc (Appendix A) were chemically synthesized and cloned into a modified pCEP4 mammalian expression vector [25]. The expression vectors were transfected into HEK293F cells (Freestyle 293-F Cells) using polyethylenimine (Sigma-Aldrich, St. Louis, MO, USA) as described previously [25]. Culture supernatants were purified by affinity chromatography using Protein A resin (GE Healthcare, Chicago, IL, USA) following the manufacturer’s instructions.

The purified recombinant proteins were separated on 4–12% NuPAGE Bis-Tris gels (Invitrogen, Carlsbad, CA, USA) and stained with Coomassie Brilliant Blue R-250 (Ameresco, Framingham, MA, USA).

### 2.3. Generation and Preparation of the Anti-CSQ VH sdAb

Three White Leghorn chickens were immunized and boosted three times with NUPR1a-His-HA-CSQ fusion protein as described previously [26,27]. Phage display chicken VH sdAb libraries were constructed using total RNA extracted from the blood, spleen, and bone marrow of immunized chickens, as described previously [26]. Genes encoding the VH fragments of chicken IgY were amplified by PCR using primers annealing to the 5′ and 3′ ends of VH genes (Appendix A) [28]. After four rounds of biopanning against CSQ-hFc, VH-displaying phages were rescued from titer plates and used for ELISA as previously described [29].

Positive clones were subcloned into a modified pET-22b expression vector (Novagen) and transformed into *E. coli* BL21 (DE3) (New England Biolabs Ipswich, MA, USA) for expression as VH-human Cκ fusion proteins (VH-Cκ) as described previously [30]. After overnight culture with IPTG induction, VH-Cκ fusion proteins in the culture supernatant were purified by affinity chromatography using KappaSelect resin (GE Healthcare Chicago, IL, USA) according to the supplier’s protocol.

### 2.4. Differential Scanning Calorimetry (DSC)

Differential scanning calorimetry (DSC) was performed by KBio Health (Seoul, Republic of Korea). PE06 VH-Cκ fusion protein was diluted to 1 mg/mL in phosphate-buffered saline (PBS, pH 7.4) and filtered through a 0.22 µm membrane prior to analysis. Thermal scans were performed from 25 °C to 100 °C at a heating rate of 1 °C/min. The melting temperature (T_m_) was determined as the peak of the heat capacity (Cp) curve.

### 2.5. Conjugation PE06 VH-Cκ Fusion Protein to HRP and Alexa Fluor 647 Fluorescent Dye

Purified PE06 VH-Cκ (1 mg/mL) was conjugated with HRP using an EZ-Link Plus Activated Peroxidase Kit, or labeled with Alexa Fluor™ 647 using the Alexa Fluor™ 647 Protein Labeling Kit (both from Thermo Fisher Scientific, Waltham, MA, USA), according to the manufacturer’s instructions. The resulting conjugates were designated PE06 VH-Cκ-HRP or PE06 VH-Cκ-AF647, respectively.

### 2.6. Enzyme-Linked Immunosorbent Assay (ELISA)

ELISA was performed as previously described with minor modifications [31]. Microtiter plates (Costar, Cambridge, MA, USA) were coated overnight at 4 °C with 100 ng/well of anti-SARS-CoV-2 VH-His-HA-CSQ fusion protein dissolved in coating buffer (0.1 M sodium bicarbonate, pH 8.6). After blocking, anti-CSQ VH-Cκ fusion proteins were added to the wells in serial dilutions and incubated for 1 h at 37 °C. Bound antibodies were detected using HRP-conjugated goat anti-human kappa light chain antibody (Thermo Fisher Scientific, Scientific, Waltham, MA USA) and 2,2′-azino-bis(3-ethylbenzothiazoline-6-sulfonic acid) (ABTS) substrate (Amresco LLC, Solon, OH, USA). Absorbance at 405 nm was measured using a SkanIt microplate reader (Thermo Fisher Scientific, USA). All experiments were performed in triplicate.

To assess the antigenic reactivity of PE06 VH-Cκ-HRP, microtiter plates were coated overnight at 4 °C with 100 ng/well of recombinant SARS-CoV-2 RBD (Sino Biological, Beijing, China). After blocking, plates were incubated with anti-SARS-CoV-2 VH-His-HA-CSQ fusion protein in serial dilutions for 1 h at 37 °C, washed with PBS containing 0.05% Tween-20 (PBST; Biosesang, Seongnam, Republic of Korea), and incubated with PE06 VH-Cκ-HRP or an anti-HA-HRP antibody (SC7392; Santa Cruz, Dallas, TX, USA) at 250 nM. Plates were washed again with PBST, and ABTS substrate was added. Absorbance at 405 nm was measured using a SkanIt microplate reader (Thermo Fishcer Scientific, Vantaa, Finland). All experiments were performed in triplicate.

### 2.7. Biolayer Interferometry (BLI)

The affinity of PE06 VH-Cκ was measured on an Octet R8 system (Sartorius, Gottinggen, Germany) according to the manufacturer’s instructions. HIS1K (anti-His tag) biosensors were equilibrated in Dulbecco’s phosphate-buffered saline (DPBS) for at least 10 min. The anti-claudin VHH–His–HA–CSQ fusion protein was loaded onto the biosensors at 166.7 nM. Association was recorded by dipping the sensors into PE06 VH-Cκ solutions at 0, 7.93, 23.8, 71.4, 238.1, and 714.3 nM for 900 s, followed by dissociation in DPBS for 1800 s. Data were analyzed using Octet Analysis Studio v13.

### 2.8. Immunoblot Analysis

Immunoblot analysis was performed as described with minor modifications [29]. Briefly, recombinant SARS-CoV-2 RBD protein was separated on 4–12% Bis-Tris gel (Thermo Fisher Scientific, USA) and transferred to a 0.45 µm nitrocellulose membrane (Bio-Rad Laboratories Inc., Hercules, CA, USA). After blocking, the membrane was probed with 200 ng of anti-SARS-CoV-2 VH-His-HA-CSQ fusion protein. Following washes with 1× Tris-buffered saline containing 0.05% Tween-20 (TBST), membranes were incubated overnight at 4 °C with 1 μg/mL MA02 VH-Cκ, SE09 VH-Cκ, PE06 VH-Cκ, or PE06 VH-Cκ-HRP. The lane incubated with PE06 VH-Cκ-HRP was washed and further incubated with an anti-His-HRP antibody (MA1-21315-HRP; Thermo Fisher Scientific, USA). After washing, membranes were developed using SuperSignal™ West Pico Chemiluminescent Substrate (Thermo Fisher Scientific, USA) and visualized with an enhanced ChemiDoc ^TM^ Touch Imaging System (Bio-Rad Laboratories Inc., Hercules, CA, USA).

### 2.9. Flow Cytometry

HEK293, HEK293-hCLDN18.2 OE, and SNU-601 cells (3 × 10^5^ cells) were resuspended in 100 μL of flow cytometry buffer (1% BSA and 0.02% sodium azide in PBS) and incubated with 250 nM anti-human claudin 18.2 VHH-His-HA-CSQ fusion protein for 1 h at 37 °C. Cells were washed twice with flow cytometry buffer and incubated with PE06 VH-Cκ-AF647 at concentrations of 100, 250, 500, or 1000 nM for 1 h at 37 °C. After washing with flow cytometry buffer, fluorescence intensity was measured on a FACS Canto II (BD Biosciences, Heidelberg, Germany) and analyzed with FlowJo software, version 10.4.0 (Tree Star, Ashland, OR, USA).

### 2.10. Protein Structure Modeling

Structural models of three sdAbs (SE09 VH-Cκ, MA02 VH-Cκ, and PE06 VH-Cκ) in complex with CSQ were generated using AlphaFold3 by submitting full-length sdAb and CSQ sequences in paired format [32]. Default settings were used without manual template selection. Model confidence was evaluated using pLDDT scores and PAE matrices. Predicted structures were analyzed in PyMOL (Schrödinger, LLC, New York, NY, USA), with a focus on antibody–antigen binding interfaces.

### 2.11. Statistical Analysis

All statistical analyses were performed using GraphPad Prism Ver.10.2.3 (GraphPad Software, Inc., La Jolla, CA, USA).

## 3. Results

### 3.1. Generation of Anti-CSQ Chicken sdAb

Chickens were immunized with human NUPR1a-His-HA-CSQ fusion protein (Figure 1). From the immunized chickens, total RNA was isolated from peripheral blood mononuclear cells, bone marrow, spleen, and bursa of Fabricius to prepare a cDNA library. From this library, gene fragments encoding VH were amplified using gene-specific primers (Appendix A). The resulting VH gene fragments were used to construct a phage-displayed chicken VH sdAb library. After four rounds of bio-panning against CSQ-hFc recombinant fusion protein immobilized on Dynebeads^TM^ M-270 Epoxy (Thermo Fisher Scientific, USA), sdAb clones reactive to the CSQ-hFc fusion protein were selected by phage ELISA. All positive clones were subjected to nucleotide sequencing, and unique clones were identified.

### 3.2. Characterization of Anti-CSQ sdAbs

For further characterization, VH genes from the 7 positive clones were subcloned into a prokaryotic expression vector and transformed into BL21(DE3) for protein expression. The resulting VH-Cκ fusion proteins were purified via affinity chromatography, and their purity was assessed by sodium-dodecyl sulfate polyacrylamid gel electrophoresis (SDS-PAGE; Appendix A).

In an ELISA, the reactivity of individual sdAbs to CSQ was tested using an ELISA plate coated with recombinant anti-SARS-CoV-2 VH-HA-CSQ fusion protein. All 7 clones displayed specific reactivity to the coated CSQ antigen, with half-maximal effective concentrations (EC_50_) ranging from 13.84 to 75.54 nM (Figure 2). For subsequent experiments, three clones-MA02 (EC_50_ = 16.47 nM), PE06 (EC_50_ = 50.60 nM), and SE09 (EC_50_ = 75.54 nM) were selected based on their EC_50_ values, expression levels, and minimal non-specific binding.

### 3.3. Thermal Stability of PE06 VH-Cκ sdAb

The thermal stability of the PE06 VH-Cκ fusion protein was evaluated using differential scanning calorimetry (DSC). The fusion protein exhibited a single unfolding transition with a melting temperature (T_m_) of 48.2 °C, indicating conformational stability comparable to that of the VH region in an IgG molecule under physiological conditions (Appendix A) [33].

### 3.4. Evaluation of Anti-CSQ sdAb in ELISA

Recombinant PE06 VH-Cκ fusion protein was tested as a secondary antibody for detecting recombinant CSQ protein in ELISA following conjugation to HRP. Microtiter plates were coated with recombinant SARS-CoV-2 RBD and blocked with BSA. Anti-SARS-CoV-2 VH-His-HA-CSQ fusion protein—derived from a single-chain variable fragment (scFv) clone (Appendix A; clone 35–78 isolated from a vaccinee) [24]—was added in serial dilutions. The amount of bound CSQ fusion protein was quantified using either PE06 VH-Cκ-HRP or an anti-HA-HRP tag antibody. The EC_50_ values of anti-SARS-CoV-2 VH-HA-CSQ fusion protein were determined to be 50.40 nM and 46.30 nM using PE06-Cκ-HRP and an anti-HA-HRP antibody, respectively (Figure 3), showing comparable results. These findings indicate that recombinant PE06 VH-Cκ fusion protein can be used to quantify bound CSQ fusion proteins—primarily antibodies—in ELISA.

### 3.5. BLI-Based Characterization of PE06 VH-Cκ Binding

The binding properties of PE06 VH–Cκ were examined by bio-layer interferometry (BLI). Anti-claudin VHH–His–HA–CSQ fusion protein was immobilized on HIS1K biosensors and exposed to PE06 VH–Cκ at concentrations ranging from 7.93 to 714.3 nM. The sensorgrams showed clear concentration-dependent binding. Kinetic analysis with Octet Analysis Studio v13 yielded an association rate constant (k_a_) of 9.43 × 10^3^ M^−1^ s^−1^ and a dissociation rate constant (K_dis_) of 3.71 × 10^−5^ s^−1^, corresponding to an equilibrium dissociation constant (Kᴅ) of 3.93 nM (Figure 4).

### 3.6. Application of Anti-CSQ adAb as Secondary Antibody in Immunoblot Analysis

PE06 VH-Cκ-HRP was also tested as a secondary antibody in immunoblot analysis. Recombinant SARS-CoV-2 RBD was subjected to SDS-PAGE, transferred to a nitrocellulose membrane, and probed with anti-SARS-CoV-2 VH-His-HA-CSQ fusion protein. The membrane was then incubated with either PE06 VH-Cκ-HRP or an anti-His-HRP antibody. After washing, membranes were developed using enhanced chemiluminescence (ECL). In a parallel experiment, membranes were incubated with recombinant PE06 VH-Cκ, SE09 VH-Cκ, or MA02 VH-Cκ fusion proteins, washed, and probed with an anti-His-HRP antibody. Following washes, membranes were developed by ECL. In both immunoblot assays, PE06 VH-Cκ-HRP, as well as PE06 VH-Cκ, SE09 VH-Cκ, and MA02 VH-Cκ fusion proteins, were detected at positions consistent with the expected molecular weight of recombinant SARS-CoV-2 RBD (60 kDa) in all lanes (Figure 5), demonstrating that PE06 VH-Cκ-HRP can serve as a secondary antibody in immunoblot analysis.

### 3.7. Flow Cytometry Analysis Using Anti-CSQ sdAb

PE06 VH-Cκ-AF647 was evaluated as a secondary antibody in flow cytometry analysis. HEK293 cells transfected with the human CLDN18.2 gene (HEK293-hCLDN18.2 OE), SNU-601 cells endogenously expressing claudin 18.2, and wild-type HEK293 cells were incubated with anti-human claudin 18.2 VHH-His-HA-CSQ fusion protein. This VHH clone was derived from an alpaca immunized with human 18.2 and identified through bio-panning. After washing, cells were incubated with PE06 VH-Cκ-AF647 at concentrations of 100, 250, 500, or 1000 nM. In a parallel experiment, cells were incubated with PE06 VH-Cκ-AF647 without prior treatment with anti-human claudin 18.2 VHH-His-HA-CSQ fusion protein. A significant fluorescence shift was observed in HEK293-hCLDN18.2 OE and SNU-601 cells sequentially incubated with anti-human claudin 18.2 VHH-CSQ fusion protein and PE06 VH-Cκ-AF647 compared with cells incubated with PE06 VH-Cκ-AF647 alone (Figure 6). No shift was observed in wild-type HEK293 cells. These results demonstrate that PE06 VH-Cκ-AF647 can be used as a secondary antibody to detect recombinant antibody-CSQ fusion proteins by flow cytometry. In a separate assay, we tested whether recombinant PE06-Cκ fusion protein could be used as a secondary antibody in combination with an anti-human Cκ-HRP (as a tertiary antibody) to detect anti-human claudin 18.2 VHH-His-HA-CSQ fusion protein bound to cells. The results were consistent with those obtained using PE06 VH-Cκ-AF647 (Appendix A).

### 3.8. In Silico Modeling of Anti-CSQ sdAbs and CSQ Complexes

To gain insight into antibody–antigen interactions, we performed structural modeling of PE06, SE09, and MA02 clones in complex with CSQ (Appendix A). SE09 and PE06 were predicted to bind a similar epitope on CSQ, with all three CDRs contributing to the interaction. Their binding orientations and interfaces were highly similar. In contrast, the MA02 clone was modeled to bind a different region of the antigen, primarily via CDR3.

## 4. Discussion

Single-domain antibodies (sdAbs) have emerged as powerful tools in both research and therapeutic applications due to their small size, high stability, and ability to target otherwise inaccessible epitopes. While camelid VHHs and shark VNARs have traditionally been the primary favored sources, it has been reported that chickens with a truncated light chain transgene can express heavy-chain-only antibodies [23]. In addition, single B cell screening of these chickens yielded antigen-reactive heavy-chain-only monoclonal antibodies after immunization. Structurally autonomous VH domains of chicken immunoglobulins are thought to be one of the mechanisms enabling this property [21,22]. Our study showed that VH domains of chicken antibodies can be displayed as sdAbs on the surface of M13 bacteriophage to generate a phage-displayed antibody library. From this library, we successfully isolated VH sdAbs against the immunized antigen, CSQ. From the initial set of 7 clones, 3 clones produced satisfactory yields of recombinant Cκ fusion proteins in *E. coli*. At least one clone, PE06, was stable enough to withstand chemical conjugation with HRP and a fluorescent dye. These conjugates were successfully applied in ELISA, immunoblot, and flow cytometry analyses to detect recombinant antibody-CSQ fusion proteins. Our findings validate that chickens can serve as a viable source for sdAb phage display libraries.

The Tm of PE06 VH-Cκ sdAb (48.2 °C) falls within the typical range for isolated VH (40–65 °C) [34] and Cκ (50 °C) domains [35]. In contrast, VHH domains often display higher thermal stability, with reported T_m_ values ranging from 60 °C to over 75 °C [36]. The reduced thermal stability observed for PE06 VH-Cκ compared to typical VHH domains may be attributed to the absence of the stabilizing VH-VL and CH1-Cκ interfaces [34,37]. In conventional antibodies, the light chain not only contributes to antigen recognition but also plays a critical role in maintaining the thermodynamic stability of the variable region through interdomain interactions [38], which generally results in IgG molecules exhibiting T_m_ values above 70 °C [39]. As previously reported [40], the thermostability of VH sdAbs can be increased by introducing framework region 3 substitutions-V37F, G44E, L45R and Y47G or F47G (Y in SE09 and PE06; F in MA02). We plan to evaluate these substitutions in future studies.

Our study also demonstrated that human VH and alpaca VHH sdAbs can be expressed as recombinant CSQ fusion proteins in *E. coli*, leveraging the exceptional solubility of CSQ [9]. These sdAbs were subsequently purified from *E. coli* cell lysates through a simple precipitation step using CaCl_2_, followed by solubilization with EDTA. The purified sdAbs were successfully applied in ELISA, immunoblot, and flow cytometry analyses in combination with our anti-CSQ sdAbs. Given the existence of a database containing chemically defined antibodies with their sequences (over 30,000 antibodies against 4400 different targets) [41], it is theoretically possible that these antibodies could be expressed and purified in the form of recombinant CSQ fusion proteins in *E. coli* and applied to various research applications. Furthermore, our PE06 clone can be expressed in *E. coli* and used as a secondary antibody in combination with commercially available anti-HA antibodies, anti-His antibodies, or anti-human Cκ antibodies labeled with various enzymes, fluorescent dyes, or biotin for a wide range of research purposes.

One limitation of the recombinant antibody-CSQ fusion system is that CSQ is naturally expressed intracellularly- at high levels in skeletal muscles, at moderate levels in skin, and at low levels in other tissues such as brain and intestines-making it difficult for our anti-CSQ antibodies to differentiate native CSQ from antibody-fused CSQ [42]. This hurdle is particularly problematic in immunohistochemical studies, where discrimination of antigens based on molecular weight differences is not feasible. We postulated that for some anti-CSQ antibodies recognizing conformational epitopes, it may be possible to discriminate between denatured native CSQ and conformationally intact CSQ in CSQ fusion antibodies during immunoblot and immunohistochemistry analyses. In immunoblot analysis, the PE06 and SE09 clones exhibited notably lower reactivity to de-natured CSQ protein on membranes compared to five other clones, including MA02 (Appendix A). In subsequent analyses, we performed in silico modeling of anti-CSQ sdAbs and their complexes with CSQ for the PE06, SE09, and MA02 clones. The epitopes of PE06 and SE09 were predicted to overlap despite substantial sequence differences, whereas the epitope of MA02 was spatially distinct from those of PE06 and SE09. We have previously shown that antibody specificity can be enhanced by altering heavy chain CDR3 sequences [43], and we are now investigating the potential to further refine the PE06 and SE09 clones to be more specific for conformationally native CSQ while minimizing reactivity to denatured CSQ. Alternatively, we will map the epitopes of the anti-CSQ antibodies, introduce point mutations at these sites in CSQ, and then engineer antibodies to recognize specifically the mutant CSQ, as previously described [43].

## Figures and Tables

**Figure 1 antibodies-14-00080-f001:**
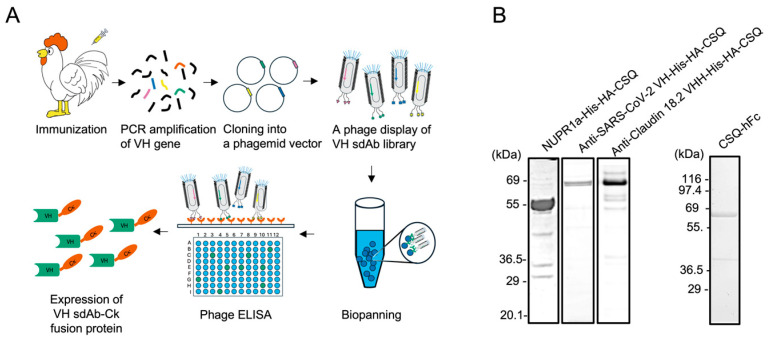
Schematic overview on the discovery of chicken heavy chain variable region (VH) only single domain antibody (sdAb) library. (**A**) Schematic presentation of chicken VH sdAb preparation: After immunization of White Leghorn chickens with the NUPR1a-CSQ fusion protein, peripheral blood mononuclear cells, spleen, bursa of Fabricius, and bone marrow were collected. mRNA was extracted from these organs and used for cDNA synthesis. Genes encoding the heavy chain variable region (VH) were obtained via PCR using gene-specific primer sets. These genes were subcloned into phagemid vectors and electroporated into *E. coli* cells. After overnight culture, a phage display of VH sdAb library was prepared and subjected to bio-panning on CSQ-hFc immobilized on magnetic bead. Positive clones were selected by phage enzyme-linked immunosorbent assay (ELISA) using microtiter plate coated with anti-SARS-CoV-2 VH-His-HA-CSQ fusion protein. The selected clones were expressed as a fusion protein with human kappa light chain constant region (Ck). (**B**) sodium dodecyl sulfate-polyacrylamide gel electrophoresis (SDS-PAGE) of recombinant proteins: NUPR1a-CSQ (~48 kDa) fusion protein, CSQ-hFc (~64 kDa) fusion protein, anti-SARS-CoV-2 VH-His-HA-CSQ (~60 kDa) fusion protein, and anti-CLDN VHH-His-HA-CSQ (~60 kDa) fusion protein were subjected to 4–12% (*w*/*v*) SDS-PAGE under reducing conditions, and the resolved proteins were visualized by staining with Coomassie Brilliant Blue.

**Figure 2 antibodies-14-00080-f002:**
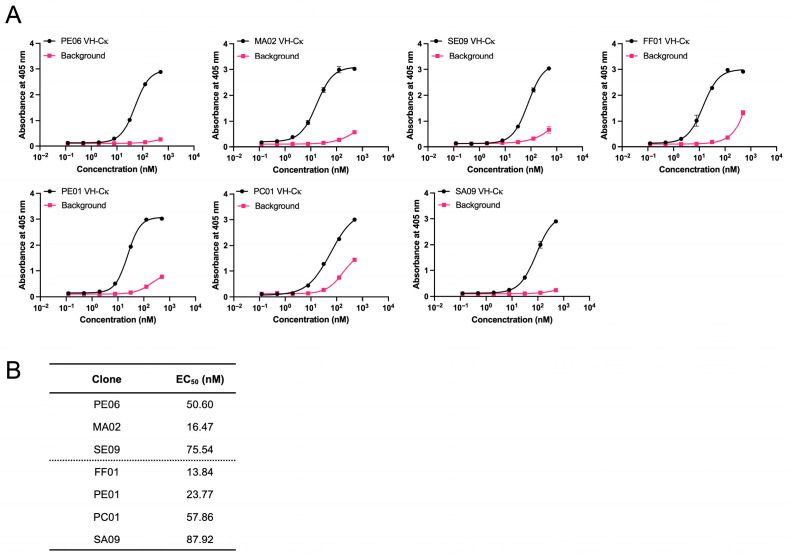
Reactivity of anti-CSQ VH clones to CSQ. Seven anti-CSQ VH single domain antibodies (sdAbs) were prepared as VH-Cκ fusion proteins using a prokaryotic expression system and subjected to enzyme-linked immunosorbent assay (ELISA). (**A**) The microtiter plate was coated with anti-SARS-CoV-2 VH-His-HA-CSQ and blocked. anti-CSQ VH-Cκ-HRP fusion proteins were added to each well of the microtiter plate. After washing, HRP-conjugated anti-human kappa light chain antibody was added. The amount of bound antibodies was determined using an ABTS (2,2′-azino-bis (3-ethylbenzothiazoline-6-sulfonic acid)) substrate solution. Optical density was measured at 405 nm. Data represent mean ± SD from triplicates. (**B**) EC_50_ values were calculated using four-parameter logistic regression in GraphPad Prism 9.

**Figure 3 antibodies-14-00080-f003:**
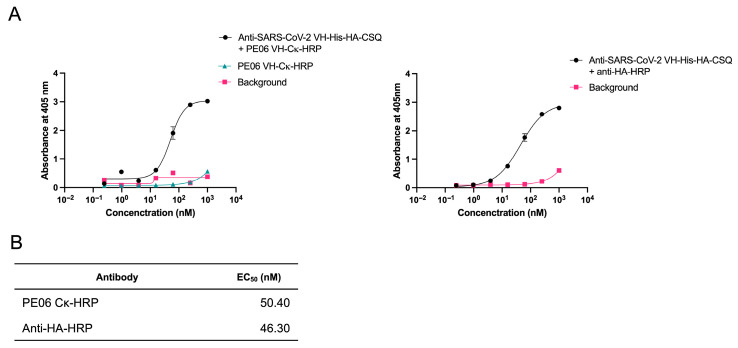
Enzyme-linked immunosorbent assay (ELISA) using horseradish peroxidase (HRP)-conjugated PE 06 VH-Ck fusion protein (PE06 VH-Ck-HRP). (**A**) PE06 VH-Ck-HRP was tested as a secondary antibody in ELISA. The wells of microtiter plate were coated with recombinant SARS-CoV-2 RBD protein. After blocking, anti-SARS-CoV-2 VH-His-HA-CSQ fusion protein was added in a serial dilution series. After washing, either PE06 VH-Ck-HRP (left panel) or an anti-HA-HRP antibody (right panel) was added to the wells. The plate was washed and incubated with an ABTS (2,2′-azino-bis (3-ethylbenzothiazoline-6-sulfonic acid)) substrate solution. Optical density was measured at 405 nm. Data represent mean ± SD from triplicates. (**B**) EC_50_ values of anti-SARS-CoV-2 VH-His-HA-CSQ fusion protein were calculated using four-parameter logistic regression in GraphPad Prism 9 using the data obtained with PE06 VH-Ck or an anti-HA-HRP.

**Figure 4 antibodies-14-00080-f004:**
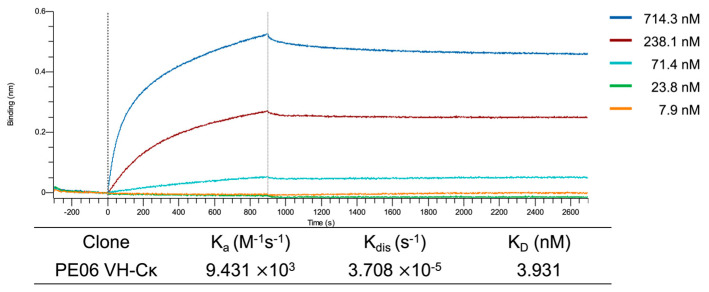
Binding kinetics of PE06 VH–Cκ to anti-claudin VHH–His–HA–CSQ fusion protein. Bio-Layer Interferometry sensorgrams recorded on an Octet R8 show the interaction of immobilized anti-claudin VHH–His–HA–CSQ fusion protein captured on HIS1K biosensors with serial dilutions of PE06 VH–Cκ. The equilibrium dissociation constant (K_D_) was determined using Octet Analysis Studio v13.

**Figure 5 antibodies-14-00080-f005:**
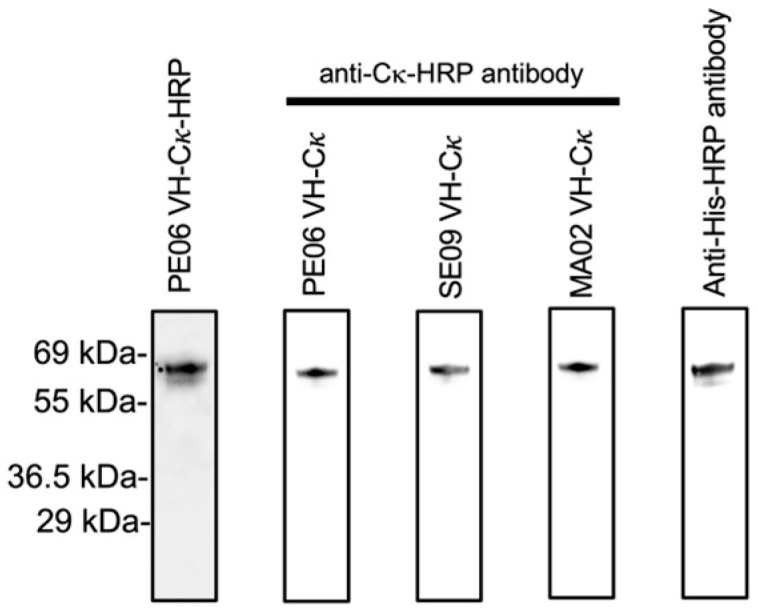
Immunoblot analysis using horseradish peroxidase (HRP)-conjugated PE 06 VH-Ck fusion protein (PE06 VH-Ck). Recombinant SARS-CoV-2 receptor binding domain (RBD) protein was separated by 4–12% sodium docecyl sulfate-polyacrylamide gel electrophoresis (SDS-PAGE) and transferred onto a nitrocellulose membrane. The membrane was probed with anti-SARS-CoV-2-VH-His-HA-CSQ fusion protein. After washing, the membrane was incubated with PE06 VH-Ck-HRP (left lane), PE06 VH-Ck, SE09 VH-Ck, or MA02 VH-Ck followed by HRP-conjugated anti-Ck antibody (three lanes in the center) or HRP-conjugated anti-His antibody (right lane). Bands were visualized using electrochemiluminescent (ECL) substrate.

**Figure 6 antibodies-14-00080-f006:**
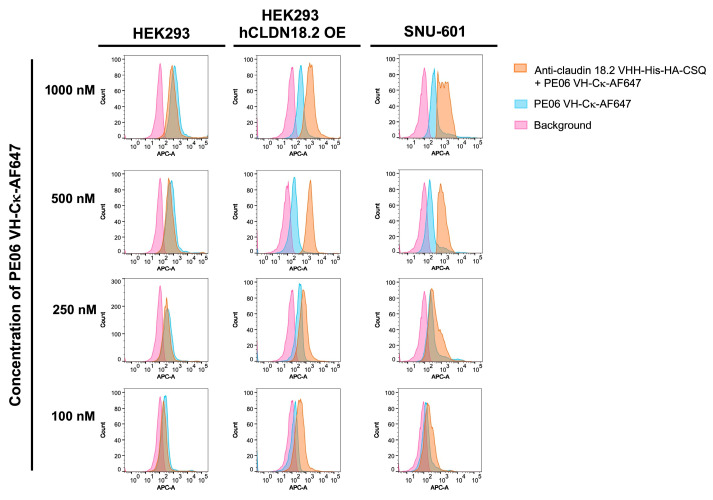
Flow cytometric analysis using Alexa Fluor 647-conjugated PE06 VH-Ck fusion protein (PE06 VH-Ck-AF647). Wild-type HEK293 cells, claudin18.2 gene-transfected HEK293 cells (HEK293-hCLDN18.2 OE) and SNU-601 cells naturally over-expressing claudin 18.2 were incubated with anti-claudin 18.2 VHH-His-HA-CSQ fusion protein at 250 nM followed by PE06 VH-Ck conjugated Alexa Fluor647 fluorescent dye (PE06 VH-Ck-AF647) in serial dilution series.

## Data Availability

The original contributions presented in this study are included in the article/Appendix A. Further inquiries can be directed to the corresponding author(s).

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
