# Peer review of "Development of a Chicken Immunoglobulin Heavy Chain Variable Region (VH) Single-Domain Antibody (sdAb) Against Calsequestrin (CSQ) and Its Application"

_2073-4468, 2025, doi:10.3390/antib14030080_

Round 1

Reviewer 1 Report

Comments and Suggestions for Authors

The paper by J. Chung et al. describes the development of a chicken immunoglobulin heavy chain variable region single domain antibody against CSQ, and the sdAb was applied to ELISA, immunoblot, and flow cytometry analyses to detect CSQ-tagged sdAbs expressed

In E. Coli. These results are very interesting to researchers in the field of antibody engineering as well as analytical chemistry.   This paper also describes the details of experiments and is well-written.  Thus, I recommend the publication in antibodies.  Aside from the recommendation, the following points should be revised before publication.

  1. I would like to know how strongly the sdAbs bind to CAQ.  Please show the Kd values.  That would be beneficial information for the readers.

Reviewer 2 Report

Comments and Suggestions for Authors

This is a well-written and interesting paper addressing an unmet need: the generation of reliable anti-calsequestrin (CSQ) detection reagents. The authors successfully establish chicken-derived VH single-domain antibodies (sdAbs) and demonstrate their application in ELISA, immunoblot, and flow cytometry. The novelty lies in leveraging chicken VH-only sdAbs for CSQ-tagged protein detection, broadening the antibody engineering toolkit. The methodology is robust, and the experiments are logically structured. However, some aspects could be clarified, expanded, or strengthened to improve the manuscript’s impact.
Major comments:
1. While the introduction highlights CSQ’s solubility-enhancing properties, it would be useful to better contextualize whyCSQ is preferable over existing tags (MBP, SUMO, GST, etc.) and what specific advantages anti-CSQ sdAbs provide compared to traditional tag-detection antibodies.

2. The discussion mentions differences in reactivity to native versus denatured CSQ, but this important limitation is not deeply explored. Can the authors clarify how this affects practical applications?

3. Some figures (ELISA curves, Western blots, flow cytometry plots) could benefit from clearer labeling, quantification, or side-by-side comparison with traditional tag antibodies (e.g., anti-His, anti-HA) to directly demonstrate advantages.

4. The Tm (48.2 °C) is lower than camelid VHHs, which may limit broader utility. Could stability be improved through engineering? This point could be discussed.

5.  Since CSQ is abundant in mammalian cells, the practicality of using anti-CSQ sdAbs in mammalian systems may be limited. More discussion on strategies to overcome this (epitope engineering, selective mutations) would strengthen the translational relevance.

Minor:
1.  There are a few typos/formatting issues (e.g., “Alexa Flour” instead of “Alexa Fluor” in some figure legends).

2. The abbreviations section is helpful, but some are redundant (e.g., listing “OE = Overexpressing”).

3. Ethical statement says no animal approval needed, but chickens were immunized. This may need clarification regarding IACUC-equivalent oversight.
